# Circadian and Immunity Cycle Talk in Cancer Destination: From Biological Aspects to In Silico Analysis

**DOI:** 10.3390/cancers14061578

**Published:** 2022-03-20

**Authors:** Mina Mirian, Amirali Hariri, Mahtasadat Yadollahi, Mohammad Kohandel

**Affiliations:** 1Department of Pharmaceutical Biotechnology, School of Pharmacy and Pharmaceutical Sciences, Isfahan University of Medical Sciences, Isfahan 8174673461, Iran; mina.mirian@pharm.mui.ac.ir (M.M.); haririamirali@gmail.com (A.H.); 2School of Pharmacology and Pharmaceutical Sciences, Isfahan University of Medical Sciences, Isfahan 8174673461, Iran; yad.mahta@gmail.com; 3Department of Applied Mathematics, University of Waterloo, Waterloo, ON N2L 3G1, Canada

**Keywords:** circadian cycle, immunotherapy, cancer immunity cycle, computational biology

## Abstract

**Simple Summary:**

The circadian cycle is a natural cycle of the body repeated every 24 h, based on a day and night rhythm, and it affects many body processes. The present article reviews the importance and role of the circadian cycle in cancer and its association with the immune system and immunotherapy drugs at the cellular and molecular levels. It also examines the genes and cellular pathways involved in both circadian and immune systems. It offers possible computational solutions to increase the effectiveness of cancer treatment concerning the circadian cycle.

**Abstract:**

Cancer is the leading cause of death and a major problem to increasing life expectancy worldwide. In recent years, various approaches such as surgery, chemotherapy, radiation, targeted therapies, and the newest pillar, immunotherapy, have been developed to treat cancer. Among key factors impacting the effectiveness of treatment, the administration of drugs based on the circadian rhythm in a person and within individuals can significantly elevate drug efficacy, reduce adverse effects, and prevent drug resistance. Circadian clocks also affect various physiological processes such as the sleep cycle, body temperature cycle, digestive and cardiovascular processes, and endocrine and immune systems. In recent years, to achieve precision patterns for drug administration using computational methods, the interaction of the effects of drugs and their cellular pathways has been considered more seriously. Integrated data-derived pathological images and genomics, transcriptomics, and proteomics analyses have provided an understanding of the molecular basis of cancer and dramatically revealed interactions between circadian and immunity cycles. Here, we describe crosstalk between the circadian cycle signaling pathway and immunity cycle in cancer and discuss how tumor microenvironment affects the influence on treatment process based on individuals’ genetic differences. Moreover, we highlight recent advances in computational modeling that pave the way for personalized immune chronotherapy.

## 1. Introduction

According to the American Cancer Society’s assessment, cancer is the leading cause of death and the major obstacle to increasing life expectancy. Cancer incidence and mortality have increased dramatically in all countries in the 21st century [1]. Common treatments for cancer include surgery, radiotherapy and chemotherapy. Currently, one of the treatments for cancer with the most growth is immunotherapy, which includes various parts; chemical and biological inhibitors and cell therapy [2].

The circadian rhythm is governed by the mammalian endogenous time system, which is synchronized with the circadian cycle of the environment produced by the Earth’s rotation. Existing data indicate that approximately 10% of the human genome is controlled by circadian clocks that also affect various physiological processes such as the sleep cycle, body temperature cycle, digestive and cardiovascular processes, and endocrine and immune systems [3].

Owing to the complex relationship between circadian rhythm, cancer and the immune system, consideration of the effect of the circadian rhythm and targeting introduces a new area in combination therapy and personalized medicine based on personal circadian rhythm. Combination therapy ameliorates effectiveness, reduces side effects, prevents drug resistance, and increases response to treatment [4,5]. The drug’s mechanism of action (MoA) and cellular pathway must be predicted to properly use two or more drugs together. Therefore, in recent years, to achieve this goal using computational methods, the interaction of the effects of drugs and their cellular pathways has been considered more strongly. In the present review, we focused on the interaction of two sets, the circadian cycle and cancer immunity, and highlighted existing methods to predict these interactions in the efficacy of immunotherapy drugs [6].

## 2. Circadian Cycle: Nature’s Clock of the Human Body

The basic structure of circadian rhythm mechanisms in living species is typically described by a self-regulating feedback loop that is autonomous from the expression of genes involved in metabolic, biosynthetic, signal transduction, and circulatory pathways. The circadian rhythms arising from primary and peripheral clocks are very similar at the cellular and molecular level [7]. The central molecular clock consists of two self-regulating transcriptional and translational feedback loops (ITTFs) that counter-modulate to create a circadian cycle of gene expression. Over 14 core and 37 associated circadian clock genes participate in the circadian clock process [7,8].

Note that the circadian locomotor output cycles kaput (CLOCK) [9] and the brain and muscle protein Arntlike1 (BMAL1) are considered to be activators of the circadian clock, while the proteins PERIOD (PER-1; PER-2; PER-3) and cryptochromes (CRY-1; CRY-2) [10] are regarded as inhibitors [8,11]. CLOCK which can be replaced by their paralogous neuronal PAS domain protein 2 (NPAS2), form a heterodimer that binds to the activator element Ebox to facilitate the transcription of target genes and modulator proteins, including PER-1, PER-2, PER-3 to activate, CRY-1 and CRY-2, thereby forming a feedback loop (Figure 1) [8,11].

Although other studies have reported that PER1, PER2, CRY1, and CRY2 play a crucial role in regulating the circadian clock rhythm, PER3 has no circadian phenotype; in the second primary cycle, the translated proteins of CRY and PER classes form heteromultimeric complexes in the cytoplasm and are transported to the cell nucleus, which negatively regulates the activation of CLOCK/BMAL1 (NPAS2) heterodimers [11,12]. These positive and negative ITTFs circulate with a circadian periodicity of approximately 24 h. In this process, casein kinase 1 (CK-1) [13] limits the negative feedback potential of PER and CRY through degeneration or phosphorylation. The ITTF, as mentioned earlier, is a CLOCK/BMAL1 heterodimer, driving the rhythmic transcription of the nuclear receptor, REV-ERB, and retinoic acid-related orphan receptors (RORs) [14] subfamilies binding to E boxes in their promotional genes. RVE/ERB-α/β and ROR-α/β can regulate BMAL1 transcription. This molecular mechanism is shared in the brain and peripheral organ systems and forms a circadian network spanning the entire body [11,12,15].

## 3. Role of Circadian Clock in Cancer; Guilty or Injured

Cancer progression processes can disrupt the circadian clock’s homeostatic equilibrium, allowing cancer characteristics to emerge more easily. Other research, also suggests that cancer is influenced by circadian rhythm disruption and that clock genes regulate it [16]. Components of the circadian cycle are known through three major mechanisms [17]:(i)Circadian clock components control the expression of hundreds or thousands of genes in various cell types, resulting in daily rhythms in different cellular functions such as nutritional metabolism [18,19,20,21], redox regulation [22,23,24,25], autophagy [26,27,28], DNA damage repair [29], protein folding [30], and cell secretion [31].

Individual equilibrium is dependent on the daily cycle of these biological processes. Changes in circadian rhythms impair these cellular activities, resulting in a cellular milieu favorable to tumor growth (e.g., metabolic reprogramming, redox imbalance, chronic inflammation, etc.). Thus, altering the circadian clock’s function or the production of clock proteins can either protect or promote cancer [32].
(ii)The cellular environment is sensed by circadian clock proteins and their associated proteins [33]. A change in the redox status of the cell, for example, can affect the CLOCK/BMAL1 complex’s affinity for DNA [34]. Furthermore, the nutritional condition of the cell or events influenced by signals determines the amounts of cofactors, including heme binding to REV-ERB [35], and the activity of factors mediating post-translational modifications, such as acetylation and phosphorylation of circadian clock proteins [36].(iii)The circadian clock controls the expression of a variety of released hormones with paracrine and endocrine activities [37]. Some of these secreted substances, such as cytokines [38,39], hormones, and neurotransmitters [40,41], can impact the clock’s activity by sending signals to their associated receptors and downstream pathways and pulling or synchronizing clocks in various tissues. These endocrine variables have the potential to serve as biomarkers for circadian function in various tissues. Furthermore, some malignancies can release excessive amounts of these hormones or cytokines that affect the circadian clock, thereby disrupting clocks in distant organs [42,43].

### 3.1. Genetic and Genomics of Circadian Cycle in Cancer

According to data from the Cancer Genome Atlas (TCGA) database, the mutation rate of clock genes in cancer is relatively low [44,45]. Despite the rarity of clock gene mutations in malignancies, studies of breast cancer survival trends have revealed that patients with mutations in all central clock genes or all *PER*s have a lower survival rate than patients without mutations [45,46]. Although gene mutations were uncommon, analyses of gene expression datasets such as TCGA and NCBI Gene Expression Omnibus (GEO) show that the circadian clock’s relative expression pattern significantly differs from that of non-tumor samples [47]. These analyzes show that a significant proportion of rhythmic transcripts (almost 15%, including *PER-1* and *CRY-1*) do not show circadian oscillations in cancer [44,45,46].

Genetic and genomic investigations of malignancies have revealed that changes in clock gene expression are more common than clock gene mutations, leading to the concept that oncogenic programs can disrupt circadian regulation, further complicating and stimulating tumor growth [45,48].

Circadian transcriptome investigations and targeted mechanistic research point to the concept that the circadian clock in normal cells exerts tight control over numerous cancer features [49,50]. Bypassing circadian regulation, these cancer traits may make it easier for normal cells to become malignant. In normal cells, regulatory systems known as checkpoints regulate DNA replication and the cell cycle [51]. Normal cells have a finite reproduction potential, whereas cancer cells are immortal and can replicate indefinitely. Cancer cells, maintain telomere length to avoid replicative senescence induced by telomere shortening. BMAL/CLOCK directly regulates telomerase expression, presumably via E-boxes in the *TERT* (Telomerase Reverse Transcriptase) gene promoter [51,52,53]. Changes in clock genes may thus have a direct impact on TERT expression, facilitating cellular immortality [52,53].

### 3.2. DNA Damage Response Pathway and Circadian Cycle

The DNA damage response pathway (DDR), DNA repair and the circadian cycle closely interact with each other. Several DNA repair genes display a circadian rhythm in their mRNA expression and protein accumulation [54]. In addition, several circadian components interact with the components of the DDR pathway. CRYs are structurally similar to photolysis, a group of enzymes involved in repairing single-stranded DNA fractures. Although mammalian CRYs lack DNA repair activity, CRY-1 facilitates ATR interaction with TIM (timeless), leading to a circadian rhythm in the ATR activity [55]. In fact, after in vivo treatment with cisplatin, which is a chemotherapy drug that causes DNA damage, MCM-2 phosphorylation affects ATR more when CRY-1 expression reaches its peak. Moreover, CRY-2 may facilitate interactions between ATR, TIM, and Chk1 [56].

### 3.3. Cell Cycle and Circadian Cycle

Many cell cycle regulators, such as cyclin-dependent kinase (CDK)2, CDK-4, CDK-6, Cdc25a, Cdc25b for cell cycle stages, various CDK inhibitors, and several cyclins, have daily cycles in their expression [57]. The CLOCK/BMAL or ROR/REV-ERB arms of the circadian oscillator control some of these genes directly [58]. The temporal control of G1/S transition factors, which regulate the G1/S transition by influencing P16-Ink4A expressions, is an example of a circadian cell cycle checkpoint. P16-Ink4A causes cell cycle arrest by blocking the CDK-4 and CDK-6-mediated phosphorylation of pRB, which sequesters E2F transcription factors into inactive pRb/E2F complexes. At the P16 promoter (*Cdkn2A*), the PER-2 clock component interacts with the NONO RNA binding protein, resulting in a daily rate of P16Ink4A expression. When the circadian rhythm is disrupted, the temporal regulation of the G1/S transition is disturbed, resulting in an enhanced proliferation [59].

### 3.4. Tumor Suppressor and Circadian Cycle

Other cell regulatory factors, such as tumor suppressors and oncogenes, demonstrate circadian modulation in addition to cell cycle regulators. P53, as one of the most important tumor suppressor proteins [59], has circadian oscillations, and its transcription, activity, and stability are regulated by BMAL1 and PER2 [60]. The lack of PER2 inhibits p53 activation in DNA damage, which could be due to the role of PER2 in p53 nuclear stability and translocation. BMAL expression is also downregulated, affecting the p53-dependent activation of p21 [61]. The stability of the protein p53 is aided by a second process. The circadian rhythm is trained by ATF4 (activator of transcription factor 4) [62]. The major ubiquitin ligase responsible for the stability of the p53 protein is p14ARF (Cdkn2A), a recognized inhibitor of MDM2. The clock genes thus exert control over p53 on multiple levels. Moreover, tumor suppression mechanisms guarantee that circadian rhythmicity is maintained. Indeed, the p53 pathway (namely, p53 and MDM2) modulates PER2 levels, forming a feedback loop that could be useful for tumor suppression and anticancer therapy [63].

In several types of cancers, the RAS/MAPK signaling pathway is particularly active. Based on experimental studies of non-cancerous tissue, RAS activity can regulate circadian rhythms [64]. The phosphatidylinositol-3-kinase (PI3K), which has pleiotropic effects on non-malignant cells and appears to be required for the heterodimerization of BMAL1/CLOCK [65], is a key component in the RAS signaling pathway. This condition could explain a recent finding that abnormal RAS activation causes changes in circadian rhythm, presumably due to BMAL1/CLOCK activity and *Ink4a/Arf* overexpression [66].

### 3.5. Inflammation, Immunosuppression and Circadian Cycle

Tumor formation is aided by abnormal inflammation, immunosuppression, and circumventing immune surveillance by boosting cell proliferation, cell survival, angiogenesis, and invasiveness. The immune system and inflammatory processes are both regulated by the circadian clock and its components. NF-κB, one of the key inflammatory regulators, is modulated by the circadian clock components BMAL1, CLOCK, and CRY [67]. During an inflammatory response linked with NF-κB activation via phosphorylation of the p65 subunit, BMAL1 is negatively regulated physiologically. CLOCK, on the other hand, binds to p65 and increases NF-κB-induced inflammatory gene transcription [68]. CRYs regulate NF-κB activation by modulating adenyl cyclase, an enzyme that regulates cAMP levels and increases protein kinase A (PKA), which promotes NF-κB activation by phosphorylating p65, resulting in a lack of CRY in fibroblasts and macrophages and decreased expression of NF-κB targets such as TNF, IL6, and CXCL1 [69].

Immunosuppression allows tumors to evade immune monitoring, which is essential for tumor development and spread. Due to abnormal expression of proinflammatory cytokines such as CXCL12, IL-18, IL-1, IL-10, and other immunosuppressive components in the tumor microenvironment, circadian clock irregularity could play a key role in immunosuppression [70]. Among the immunosuppressive cytokines, the transforming growth factor (TGF) is a key mediator. TGF-β slows dendritic cell development, alters the balance from type 1 helper T cells (Th1) to Th2, decreases the activation of cytotoxic T lymphocytes, and increases regulatory T cells (Tregs) [71]. TGF-β signaling has been linked to components of the circadian clock in a variety of physiological situations [72]. TGF-β mRNA expression fluctuated with circadian rhythms under the control of CLOCK-BMAL1 heterodimers. An animal study has shown that a lack of the *Clock* gene in mice increases TGF-β expression [73].

According to a recent study, in metastatic melanoma, high BMAL1 levels are related to enhanced T cell infiltration and activation. Furthermore, patients with high BMAL1 expression respond to anti-PD-1 therapy better than those with low BMAL1 levels [74]. Immune cell development, trafficking, and function are all influenced by circadian clock genes. BMAL1 orchestrates rhythmic oscillations in the number of white blood cells in non-cancerous situations and is known to play a function in B-cell development with CRY [75]. REV-ERBs play a role in macrophage cell motility and adhesion. *Clock* deficiency causes mice to have a low number of Th1 cells. RORt and NFIL3 regulate the differentiation of IL17-producing CD4 + (Th17) helper T cells through BMAL1, CLOCK, and REV-ERB [75].

## 4. Cancer Immunity System and Tumor Microenvironment

In recent years, immunotherapy has become a revolutionary treatment with mild side effects and a higher survival rate in most malignancies.

The tumor microenvironment (TME) consists of several types of cells, including tumor cells, fibroblasts, stromal cells, immune and inflammatory cells, and extracellular matrix (ECM) [76]. Immune cells in TME are crucial factors influencing tumor growth and response to therapy [76,77]. It is believed that many important immune cells are involved in the cancer process, including natural killer cells (NK) [78], tumor-associated macrophages (TAM) [79], natural neutrophil killer T (NKT) cells, dendritic cells (DCs), T lymphocytes, and B lymphocytes [80]. A growing body of evidence strongly supports the idea that immune disorders in patients can be applied to predict the outcome of various solid cancers, such as non-small cell lung cancer (NSCLC) and liver cancer. The consensus indicated that high macrophage infiltration in TME could show the rate of therapy resistance and prediction of prognosis [81]. Regarding effects, several early clinical studies targeting macrophages were successfully completed [82]. The activation of numerous T cells has been suppressed using immunosuppressive TME factors, which ultimately facilitates the tumor cells’ escape from immune systems, thus promoting cancer progression and metastasis. T cells are significantly associated with the prognosis of NSCLC [83,84].

Stimulation of immunity against cancer can be obtained through a circulatory mechanism called the cancer immunity cycle (See Figure 2). In the first step, antigen-presenting cells (APCs) can efficiently take up and process neoantigens from dying cells, including presenting target antigens and tumor cells [85]. To generate an immune response against tumor cells, they must be accompanied by cytokine release and activation of signals to induce peripheral tolerance against tumor antigens. Then, DCs present tumor-associated antigens (TAAs) to the T cells. The effector T cells are then activated to eliminate cancer-specific antigens. With the release of chemokines, effector lymphocytes activate, transfer, and infiltrate tumor tissue. Subsequently, cytotoxic lymphocytes (CTL) recognize malignant cells and associate them with them. Eventually, the cancer cells die. Furthermore, additional antigens are released enhancing the tumor’s immune response in subsequent cycles. Here, understanding how the circadian clock modulates immunity is critical [86].

### Cancer Immunity Cycle

A number of sequential actions must be initiated and allowed to proceed and repeatedly expand for an anticancer immune response to lead to the efficient destruction of cancer cells [87,88]. These processes are known as the cancer immunity cycle. Neoantigens produced by oncogenesis are released and collected by dendritic cells (DCs) in the first step (Step 1). If this step is to result in an anticancer T cell response, it must be followed by signals that define immunity, otherwise peripheral tolerance to tumor antigens will be generated. Proinflammatory cytokines and substances generated by dying tumor cells or the gut microbiota could be examples of immunogenic signals [87,89].

In the second step (Step 2), the collected antigens on MHCI and MHCII molecules are then presented to T cells, resulting in the priming and activation of effector T cell responses against cancer-specific antigens that are recognized as foreign or for which central tolerance has been broken (Step 3) [87,89].

At this point, the nature of the immune response is defined with a vital balance indicating the ratio of T effector cells to T regulatory cells being crucial to the final result. In step 4, the activated effector T cells migrate to the tumor bed and eventually infiltrate it in the next step (Step 5), where they selectively recognize and connect to cancer cells via the interaction of their T cell receptor (TCR) and the corresponding antigen coupled to MHCI (Step 6), and destroy their target cancer cell (Step 7) [87,89]. Once again, the cancer cell is killed (Step 1), releasing additional tumor-associated antigens (Step 1) that expand the range and depth of the response. The cancer-immunity cycle does not work properly in cancer patients. Many of the variables restricting the production of effector cells are found in the tumor microenvironment, such as the presence of tumor antigens that cannot be recognized, DCs misidentifying antigens as self rather than foreign, and T cells that are unable to infiltrate the tumors adequately [87,88,89].

## 5. Circadian and Immunity Cross Talk

The circadian clock acts as a gate, controlling many aspects of cancer-related immune functions, including the release and presentation of antigens to cancer cells, initiation and activation of immune effector cells, trade and infiltration, immunity to tumors, and elimination of tumor cells (Figure 3).

### 5.1. Steps 1–3: Antigen-Presentation and T Cell Activation

Cancer cells usually contain TAAs and tumor-specific antigens (TSA). After the tumor cells’ death, their antigens are released into the plasma and then, captured by DCs and macrophages. These cells are potent antigen-presenting cells (APCs) in the human body. When these are stimulated by inflammatory antigens or cytokines, particularly IL1β, TNFα, and TGFβ, immature DCs can differentiate into mature DCs and express MHC II molecules on their surface [90].

TAAs and TSAs are commonly found in cancer cells. DCs and macrophages trap these antigens that are released into the plasma when tumor cells die. Antigen-presenting cells (APCs) in the human body include DCs, macrophages, and B cells [91]. Various inflammatory cytokines, such as IL-1, TNF-α, and TGF-β, can induce the maturation of immature DCs, which then express MHC II molecules on their surface [92]. Co-stimulatory molecules and adhesion molecules are both expressed at higher quantities on their surface at the same time. DCs can also release IL-12, which induces T cells in tumor microenvironments (TMEs) to develop into T helper (Th) cells to aid cell eradication and growth suppression [93]. Moreover, the expression levels of co-stimulatory and adhesion molecules are significantly increased on their surface. Therefore, tumor antigens are extracted from peripheral tissues to trigger and activate T cells.

In addition, DCs can secrete IL-12, which differentiates TME T cells into T helper cells (Th) to promote the elimination of tumor cells and the inhibition of growth [94]. Functional molecules of the circadian cycle such as CLOCK, PER, and BMAL1 exhibit daily oscillations and indicate that DC hosts are subject to circadian regulation [95].

Previous studies showed that the expression of proinflammatory cytokines, co-stimulatory molecules, and MHC II in DC was elevated in *Rev-Erbα*- and *Rev-Erbβ*-deficient mice. This finding demonstrated that Rev-Erbs negatively modulated the DC development and activation and were important in antigen presentation [96]. These studies prove that clock genes can regulate DC functions to some extent. However, the magnitude of the effects and mechanisms involved needs to be further explored.

TAMs are separated into two polarized extremes in this broad spectrum of activation states, notably M1-type TAM macrophages (classically active TAMs) and TAM Type M2 (alternatively activated anti-inflammatory). They exert integrated effects as central regulators of the TME complex, recruitment and activation of T helper (Th) cells through releasing a wide various cytokines [97]. M1-type TAMs also play a role in pathogen-associated molecular models (PAMPAs), causing APC maturation and affecting tumor cells directly. In addition, the dominant-negative expression of CLOCK, REV-ERB, or ROR, results in fewer Th17 cells in mice [98]. RORt is required for the formation, differentiation, and survival of effector subsets of T cells, which produce the cytokines IL-17A, IL-17F, GMCSF, and IL-22, as well as Chemokine (C-C motif) ligand 20 (CCL20) and have anticancer capabilities in animals [99].

### 5.2. Steps 4–5: Lymphocyte Transportation and Infiltration

Lymphocyte transportation and localization play an antitumor role, involving adhesion chemokines and co-stimulatory molecules. Rhythmic fluctuations in the expression of adhesion molecules have been observed in tissues, including intercellular adhesion molecule 1 (ICAM-1) and 2 (ICAM-2), and vascular cell adhesion molecule 1 (VCAM-1) [100].

Other analyzes have revealed that *BMAL11* (a circadian clock gene) regulates the expression of ICAM1 and VCAM1 in endothelial cells or leukocyte subsets [101]. Studies have shown that T and B cell migration exhibit stronger discontinuous oscillations in the lymph nodes, compared to the thymus and bone marrow. This phenomenon is closely associated with the expression of CXCR-4, CCL-20 and CX3CR-1, which are controlled by glucocorticoids, catecholamines, and the hypoxia-inducible factor 1α (HIF-1α) signaling pathway [101,102]. Autonomous cell clocks are essential to lymphocyte production and critical factors in the migration of T and B cells [101].

The central circadian clock gene plays a crucial role in regulating tumor immunity by upregulating OLFML3 transcription in glioblastoma, which recruits immunosuppressive microglia into the tumor microenvironment [103]. These studies suggest that the lymphocyte transport subset is strongly under the control of components of the cellular circadian clock; therefore, optimizing the application of tumor immunotherapy is critical [103]. Regarding the infiltration of lymphocytes in TME, the circadian clock genes in clear cell renal carcinoma (ccRCC), including *CLOCK*, *BMAL1*, and *CRY-1* have been positively correlated with a variety of immune cell infiltrates such as neutrophil cells, DCs, and CD4+ T cells [101]. A similar phenomenon was also observed in cancer patients: CLOCK and the BMAL1 were closely associated with the extent of CD8+ T cell infiltration. There is evidence that compared to *BMAL1* expression in patients with healthy skin, *BMAL1* expression in melanoma patients is significantly changed, which represents a dysfunctional circadian clock and correlates positively with T-cell infiltration/activation. However, these studies focus on the level of transcription of the clock genes and therefore do not reflect extensive changes. The mechanism by which the circadian clock modulates the immune infiltration of tumors remains to be explored [104].

### 5.3. Steps 6–7: Recognition and Elimination of Cancer Cells

When the immune system fails to respond to checkpoint blockade therapy, it is generally regarded as a poor prognostic indicator. Cancer antigen-specific T and NK cells can be manipulated by circadian clock components to destroy cancer cells. Recent studies have indicated that the circadian clock and immunological escape are linked to each other. In metastatic melanomas, the expression of *BMAL1* is responsible for T-cell activation, differentiation and exhaustion markers, CTLA-4, PD-1, and PD-L1 [105,106]. Many immune-related pathways, including the PD-L1 and PD-1 checkpoint pathway in cancer, as well as the T cell receptor signaling pathway and TNFα signaling pathway, were found to be enriched in circadian clock pathways [44]. Circadian clock genes *Per1* and *Cry2* are connected to the expression of CD4+ T cells and the expression of PD-1 in normal lung tissues. Furthermore, the percentage of PD-1+ Type 17 cells and PD-1 levels on individual cells are decreased when Ror is knocked down. Cancer treatment may benefit from targeting circadian clock genes [107].

There are different surface receptors on the NK cells that can activate or inhibit the growth and spread of tumor cells. Interferon-γ (IFN-γ), TNF-α, and granulocyte-macrophage colony-stimulating factor (GM-CSF) are some of the cytokines and growth factors secreted by NK cells to kill cancer cells [108]. Researchers found that when *Per2* or *Bmal1* was omitted, the NK cell’s production of IFN-γ, granzyme B and perforin was markedly reduced [109].

## 6. Cancer Immunotherapy

Effective immunotherapy against cancer has been demonstrated in clinical studies as well as experimental and preclinical studies. There are many different types of immunotherapeutic agents approved by the Food and Drug Administration (FDA) for cancer treatment. Recombinant human IL2 for the treatment of kidney cancer cells, a first monoclonal antibody for malignant B-cell tumors, the first DC-based cancer vaccine for the treatment of prostate cancer, cell therapy developed for the chimeric antigen receptor (CAR) for B-cell lymphoma, and programmed death ligand1 (PD-L1) immune checkpoint inhibitors for melanoma are examples of immunotherapies approved for the treatment of cancer over the past three decades [110].

### Immune Checkpoint Inhibitor; Good but Not Great

Undoubtedly, the most significant achievement in cancer treatment over the past decade has been the introduction of immunomodulators targeting T cells. The best inhibitory immune checkpoints mentioned in various studies are the T4 cell-associated cytotoxic molecule (CTLA-4), the programmed cell death receptor 1 (PD-1), and the programmed cell death receptor ligand 1 (PD-L1). CTLA-4 is a molecule upregulated on the surface of active T cells to prevent over-stimulation by T cell receptors (TCR) [111]. CTLA-4 competes with CD28, a TCR co-stimulatory receptor, to bind ligands such as B71 and B72; This process prevents CD28-mediated T cell activation. PD-1 is also upregulated on activated T cells by binding to its ligand, PD-L1, and transmits a negative co-stimulatory signal limiting T cell activation [112]. The oncogenic and immunosuppressive phenotype of TME is characterized by overexpression of PD-L1 by tumor cells as well as overexpression of PD-1 and CTLA-4 by T lymphocytes. Blocking these molecules triggers an immune-mediated antitumor response [113].

Ipilimumab, the first immune checkpoint inhibitor blocking CTLA-4, was approved in 2011. After that, PD-1 inhibitors such as pembrolizumab, nivolumab, cemiplimab and PD-L1 inhibitors such as avelumab, atezolizumab, durvalumab were developed for various types of cancers. They improved the results of the treatment, and even after stopping the treatment, a stable response was observed. However, their effectiveness is limited to a small number of patients.

PD-L1 is generally expressed by macrophages, specific activated T and B cells, DCs, and specific epithelial cells, particularly under inflammatory conditions. Moreover, PD-L1 is expressed by tumor cells as an “adaptive immune mechanism” to avoid antitumor responses [114]. In ovarian cancer cells, IFN-γ has been proven to cause upregulation of PD-L1, which is responsible for disease progression, while inhibition of the IFN-γ 1 receptor may reduce PD-L1 expression in the mouse models of acute myeloid leukemia via extracellular signal-regulated MEK/kinase (ERK) and MYD-88/TRAF-6 pathways. IFNγ induces protein kinase D isoform- 2 (PKD-2), essential for the regulation of PD-L1. Inhibition of PKD2 activity inhibits PD-L1 expression and promotes a potent antitumor immune response [115]. NK cells secreted IFN-γ by Janus kinase [116] 1, JAK-2 and signal transducer and activator of transcription pathways (STAT) 1, increasing the expression of PD-L1 on the surface of tumor cells. Studies on melanoma cells have indicated that IFN-γ secreted by T lymphocytes via the JAK-1/JAK-2-STAT-1/STAT-2/STAT-3-IRF-1 pathway can regulate PDL1. NK cells and T cells appear to secrete IFNγ, which induces the expression of PD-L1 on the surface of target cells, including cancer cells [115].

As mentioned, a large number of metabolic pathways affect PD-1 and PD-L1 pathways, which can also affect the response to immunotherapy drugs [117]. Additionally, the circadian cycle in tumors has a considerable effect on metabolic pathways; therefore, the effect of the circadian cycle on the response to immunotherapy drugs is significant.

## 7. Circadian Cycle and Immunotherapy; New Insight into Cancer Therapy

Owing to the rapid development of immunotherapy drugs as well as the increase in studies into the effects of circadian cycles on the prognosis and treatment of cancer, the study of the effects of these two areas has become indispensable. Currently, the application of the circadian clock in cancer immunotherapy mainly involves two aspects: drug development for biological clock targets aimed at combination therapy and chemoimmunotherapy; see Figure 4.

### 7.1. Chronotherapy; a New Way in Precision Medicine Area

Cancer chronotherapy has gained traction as a cutting-edge treatment method that takes advantage of circadian rhythms and time when administering anticancer medications and minimizes side effects on healthy cells [118,119]. Antitumor medication tolerability is significantly influenced by the timing of administration in at least 22 clinical trials [120].

In one of those clinical trials, patients with metastatic colorectal cancer (mCRC) received chemotherapy drugs (5-fluorouracil and oxaliplatin) and two randomized phase III clinical studies, which found that cancer chronotherapy reduced treatment side effects by up to five times and virtually doubled the antitumor efficacy [121]. When these two investigations were coupled with another clinical study in which 564 individuals received identical medications (497 men and 345 women in total), a meta-analysis found that the chrono-modulated pharmacological modality considerably boosted the efficacy and survival in males while diminishing those in women [122]. A clinical investigation of 199 mCRC patients treated with oxaliplatin (infusion peak 4 pm), 5-fluorouracil (infusion peak 4 am), and irinotecan given at six distinct circadian timings confirmed this sex-specificity. Both investigations found that females had a larger circadian amplitude than males and indicated a significant difference in the optimal timing between the two sexes. It has recently been demonstrated that the timing of midsleep, the circadian maximum in skin surface temperature, and physical activity differed by up to 12 h among individuals being monitored for circadian biomarkers [123].

Chronoimmunotherapy is a subcategory of cancer chronotherapy aimed at treating cancers based on specific times during the circadian cycle [124]. Some anticancer treatments, according to experimental immunotherapies, are intended to lessen the drug toxicity and increase the tumor response rate, and response length. IFN-γ, a pleiotropic cytokine important for immune system control, was found to have a more effective anticancer impact in tumor-bearing mice in the early light phase than in the early dark phase [125]. IL-2 chronotherapy, which was used to treat metastatic renal cell cancer in phase I/II clinical research, exhibited moderate toxicity, practicality in a standard care unit, and antitumor activity [126].

In another clinical study, the relationship between the effectiveness of immune checkpoint inhibitors and their infusion time during the day was investigated. In one study on 209 patients with melanoma (Stage IV), the results showed that adaptive immune responses were less strong during the initial stimulation in the evening than when stimulated during the day [5]. Although prospective studies on the timing of injections of immune checkpoint inhibitors are essential, efforts to schedule injections before mid-afternoon can be considered in the management of advanced multidisciplinary melanoma [5].

These findings could aid in the development of well-thought-out pharmacological therapies to control of circadian clock components that may be disrupted in malignancies. However, chrono-immunotherapies are still in their early stages of development, and further research into the mechanisms is needed to improve the current anticancer treatment. A crucial way to develop cancer chronotherapy is to assess the link between clock genes and existing drug targets. Prescription of a clock-related drug following the time of day may significantly improve the efficacy and reduce the side effect.

### 7.2. Combination Therapy; Circadian Targets Beside the Immune Checkpoint

There is circadian expression of PD-L1 in healthy tissue [127]. However, tumors can express the protein on all cells in an intrinsically immune-resistant manner, or they can be adaptively stimulated to express PD-L1 by nearby lymphocytes [128]. Increased expression of circadian cycle genes has been linked to an increase in the levels of immune checkpoint and effector cells in different cancers. Many different cancers were found to directly affect PD-L1 expression by downregulating the circadian cycle gene expression, which in turn has an immunosuppressive effect on tumor phenotype [44]. The type of cancer affects the nature and direction of these connections, suggesting that the disease and the environment have an impact on these influences [44].

As powerful ROR synthetic agonists, LYC-53772 and LYC-54143 may promote Th17 cell differentiation, block Treg-driven immunosuppression, and significantly increase secreted cytokines such IL-17A, IL-17F, and GM-CSF, as well as IL-22, they may have antitumor action. T cells are also resistant to PD-L1 suppression when treated with ROR agonists, which is important in reducing antitumor immunity [129]. Furthermore, when administered during ex vivo expansion, ROR agonists increase the tumor-eliminating activity of cytotoxic Th17 cells and CAR-T cells, as well as the cytotoxic activity of human T cells, allowing tumors in tumor-bearing animals to be regressed. When reactivated against a number of tumor cell lines expressing mesothelin, the function of CAR Type 17 cells was raised compared to ROR agonist-untreated cells, and they released more cytokines, including IL17A and IFN [129].

Experimental studies on mice bearing melanoma tumors have indicated that ROR agonist-primed cells possess a stem-like memory phenotype and provide long-term protection against tumor challenge when co-infused with TRP-1 Th17 and pmel-1 Tc17. In addition, the ROR synthetic agonist SR1078 has been shown to significantly boost CD8+ T cell effector responses to anticancer immunity [130].

In one study on the breast cancer cell line, Per2 silencing successfully sensitizes doxorubicin-resistant MDA-MB-231 breast cancer cells to its lethal effects [131]. In another study, when mice with Lewis lung cancer were given doxorubicin, the expression of F4/80 and CD11c in tumor tissues, and the expression of circadian genes such *Bmal1*, *Clock*, *Rev-Erb*, and *Per-2*, as well as NF-κB and IL-6 in intraperitoneal macrophages, changed significantly [107].

## 8. System Biology and Drug Research; System Pharmacology

As mentioned, using the immune system to control tumor growth and evading the immune system’s destruction has become a characteristic of cancer pathology. The relationship between the immune system and tumor formation has recently been clarified due to advances in cancer immunotherapy research and development. Immunocompromised individuals are more likely to develop cancer, and the presence of T lymphocytes in the tumor is a strong predictor of survival. In addition, therapies that activate or engage the immune system, such as cytokines, therapeutic vaccines, immune checkpoint inhibitors and monoclonal antibodies, have been shown to be effective in treating cancer [132,133].

The ability to produce a more comprehensive understanding of biological processes and their interactions with disease or medication response is one possible benefit of systems biology approaches [134]. Many stages of drug discovery and development can benefit from data-driven decision making. The results of these methodologies may make it easier to lower risk or reprioritize medicinal candidates in development, increasing their chances of success in clinical trials. Regulatory bodies, such as the US Food and Drug Administration, support the use of such methodologies because they can help enhance and streamline the evaluation of therapy safety and effectiveness [135].

To date, the most common uses of omics and systems biology have been in cancer, where describing the tumor’s molecular complexity and heterogeneity is crucial not only for the development of targeted therapeutics but also for the identification of responder patient populations [134]. The earliest effective instances of the utility of gene-expression profiling for illness subtyping, disease prognosis, and multi-gene diagnostic tests came from oncology research. In the domains of autoimmune and infectious illnesses, where systems biology is being used to characterize disease etiology, early disease diagnosis, and understanding the MoA of vaccines and adjuvants, significant progress is also being achieved. The application of systems biology techniques has enhanced the trajectory of treatment development in the above fields, and it has the potential to be effective in the realm of cancer immunotherapy in similar ways [136].

Systems pharmacology is a branch of system biology that studies the pharmacology and drug discovery area (Figure 5): (1) Drug-target interaction predictions; (2) Drug adverse effects investigations [58]; (3) Drug repositioning; and (4) Drug combination prediction are four recent achievements in system pharmacology.

### 8.1. Polypharmacology and Drug-Target Interactions Predictions

Polypharmacology, the study of how medications interact with several molecular targets, has received prominence in recent years. Polypharmacology is certainly necessary to properly comprehend the activities of medicines. The pharmacological profile of a drug can be predicted using a computational technique developed by Keiser et al. Since each target is defined by its set of known ligands, this “similarity ensemble technique” identifies each target by its chemical structure, searches for pharmaceuticals with similar chemical structure, and then predicts novel drug-target relationships [137].

Park et al. created new tools for drug repositioning using the polypharmacology method. Polypharmacology, or the fact that a medication has numerous targets, is not taken into account in these methodologies, which leaves open the question of how possible pharmaceuticals are selected from among signaling subparts that allow for their selection. PATHOME-Drug, based on drug-associated transcriptomes, was created as a subpathway-based polypharmacology drug repositioning technique. Specific to phenotypic changes (e.g., illness status changes), this tool locates subparts of the signaling cascade and identifies approved medications so that the many targets are enriched in the subparts of this tool [138].

A study by Bratsun and colleagues assessed how circadian rhythm disruption affects cell activity in the epithelial tissue as well as malignant state occurrence in a multi-scale model of tumor growth generated by circadian rhythm disruption. Gene regulation and cell dynamics are integrated into their model. Consequently, future possibilities for this modeling could include: (i) simulating targeted pharmacological therapy for various types of malignancies; and (ii) expanding the model to examine other pathways that influence tumor growth [139].

An ex-vivo human system and computational models were used in another study to analyze nivolumab’s response kinetics in head and neck squamous cell carcinoma (HNSCC) samples (N = 50) from patients with PD-1-positive tumors. They demonstrated that drug-induced variation stratifies samples by Th1-related pathways using biological assays. They built a mathematical framework and a network of systems biology to simulate and estimate antitumor phenotypes, which implicates a dynamic involvement in the induction of Th1-related cytokines and a proliferation pattern of T cells [140].

### 8.2. Drug Adverse Effects Investigations

The pharmaceutical industry faces a tremendous issue in accurately predicting the safety and toxicity of medications in the early stages of research pipelines. The integration of biological data with systems biology methodologies could lead to a significant shift in the way medication candidates are evaluated. It is possible to forecast the action of marketed medications on unexpected “side-effect” targets by using the similarity ensemble approach, which evaluates whether a drug is bound to the target based on the chemical properties it shares with known ligands [141]. Approximately half of their hypotheses were tested and found to be correct. New off-targets that explained side effects better than any known target of a given drug were prioritized using an association measure, generating a drug-target-adverse reaction network. According to current research, researchers have developed an algorithm to predict and describe proteins associated with pharmacological side effects using a large-scale analysis [141,142]. Clinical data and known drug-target interactions were combined to uncover overrepresented protein–side effect linkages. Many complex pharmacological side effects can be explained by a single protein, according to the findings. An in silico chemical–protein interactome has been created that replicates the interactions between medications known to induce at least one major side effect and a panel of human proteins. It has been discovered that drugs with similar side effects have similar chemical–protein interaction patterns [142]. Their research has uncovered the molecular basis of various adverse events by looking into the links between drugs and off-target effects. Other research has been undertaken to successfully anticipate pharmacological side effects by combining systems biology with structural or chemoinformatics analyses [142].

Pharmacovigilance and omics data were combined in a study by Ling Han and colleagues to examine the relationship between multi-omics variables and immune-associated adverse events, providing risk ratios across different cancer types. These researchers have discovered an LCP1 and ADPGK two-variate regression model that reliably predicts immune-related adverse outcomes. An independent cohort of patients further validated the biomarkers LCP1 and ADPGK. Combining pharmacovigilance and multi-omics data, they uncovered prospective indicators of immune-related side effects in cancer immunotherapy [143].

Similarly, in another study, Relógio et al. used a unique mathematical model of irinotecan cellular pharmacokinetics and dynamics connected to a representation of the core clock to predict treatment toxicity in a model of colorectal cancer (CRC) cellular metabolism. Two human colorectal cancer cell lines serve as an in vitro experimental system for the progression of human colorectal cancer, and the mathematical model is adjusted to account for all three possible scenarios. Quantitative data sets and timing-dependent data on cell death are accurately predicted by their model, which agrees with the data. As a result, treatment can be tailored to each cell’s internal clock, according to the model’s time-dependent toxicity predictions. According to their findings, toxicity may be affected by the time-dependent degradation of the drug-activating protein and an oscillation in the death rate [144].

### 8.3. Drug Repositioning

An alternative to drug discovery is to seek new therapeutic uses for existing medications through drug repositioning or repurposing. One of the primary benefits of drug repositioning is that it can speed up clinical trials for relocated drugs by cutting down on the risks associated with drug development. Iorio et al. devised a method that uses the similarity in molecular activity signatures of all medications to compute pair-wise similarities in all drugs’ effects and mechanisms of action [145]. Drugs were grouped into a network based on the scores they received. The notion of networks was then used to divide medications into groups of densely connected nodes (i.e., communities). Many of the chemicals found in these drug communities have the exact underlying mechanism of action and often have the same targets and pathways [146]. When medications are colocated in networks, this suggests that they have similar molecular action to other drugs in the same community, which helps to identify repositioning opportunities. The “PREDICT” algorithm, proposed by Gottlieb et al., can handle both authorized medications and new substances. It is based on the notion that comparable medications are used to treat similar conditions and uses the chemical similarity of drugs and disease–disease similarity measurements to forecast new drug indications [147]. A number of in-silico medication repositioning techniques using gene expression data have also been published.

A new deep learning model, Cancer Drug Response profile scan (CDRscan), has been developed by Yoosup Chang and colleagues to predict anticancer drug responsiveness based on the genomic profiles of 787 human cancer cell lines and the structural profiles of 244 drugs from a large-scale drug screening assay. The genomic mutational fingerprints of cell lines and the molecular fingerprints of pharmaceuticals are processed separately by ‘virtual docking’, an in silico drug treatment model, in CDRscan’s two-step convolution architecture. CDRscan has a high prediction accuracy (R2 > 0.84; AUROC > 0.98) based on the goodness of fit between observed and predicted drug responses. 14 oncology medications and 23 non-oncology drugs were found to have novel cancer indications after CDRscan was applied to 1487 approved drugs. CDRscan is predicted to be able to identify the most effective anticancer treatments for each patient’s genetic profile through further clinical validation [148].

### 8.4. Drug Combination Predictions

In order to get the most out of a single medicine, it is vital to use combination therapies that affect numerous targets at once. Systems biology techniques have been used to understand and forecast possible medication combinations [149]. In order to mimic the effects of medication combinations and produce experimentally testable interventions, computational methodologies leveraging dynamic modeling have already been applied. A lack of biochemical reaction kinetics means that these dynamic models can only be used to study the mechanism of action of medication combinations at this time [150].

Kohandel and colleagues created the model to predict patient response to anti-PD-1 immunotherapy and improve the rate at which patients responded. In this way, they find biomarkers of patient response and probable pathways of medication resistance. Systems biology informed neural networks (SBINNs) are developed by the researchers to calculate patient-specific kinetic parameter values and to forecast clinical outcomes. They demonstrate how transfer learning can increase the SBINN’s response prediction accuracy using simulated clinical data. IL-6 inhibition, recombinant IL-12, and anti-PD-1 immunotherapy are all part of the triple combination therapy that these researchers are developing in order to get the best possible results for their patients. Protein expression levels vary significantly across response phenotypes, which is consistent with recent clinical results [151].

## 9. Conclusions

Many questions about the effect of circadian rhythms on cancer cells and immune cells remain unanswered, and more fundamental studies are needed. However, many clinical studies and bioinformatics modeling have demonstrated the effect of circadian rhythm on the therapeutic effects of drugs.

Moreover, immunotherapy is one of the mainstays of cancer treatment today. Various studies are underway to increase the effectiveness or decrease drug resistance in the use of combination therapy. Thus, the investigation of the relationship between immunotherapy drugs and circadian cycles can generate a new field in combination therapy and personalized medicine.

## Figures and Tables

**Figure 1 cancers-14-01578-f001:**
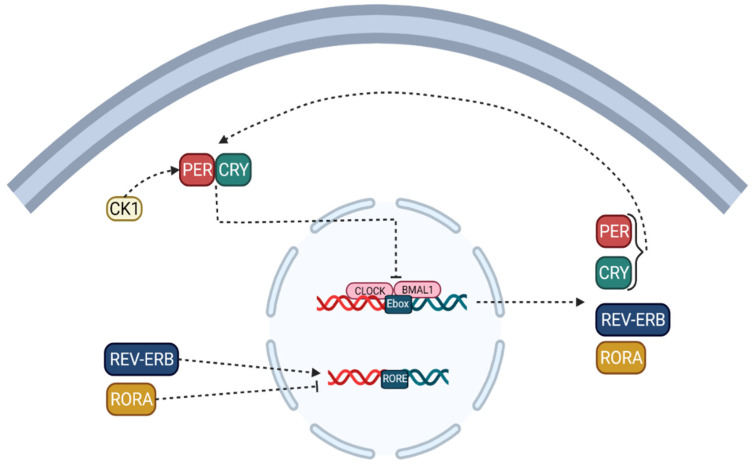
Intracellular circadian cycle pathway. The accumulating PER and CRY proteins bind to CLOCK/BMAL1 and switch them from an activated state to an inhibited state, blocking the transcriptional activity of downstream genes. ROR/REV-ERB regulates the main feedback loop by acting on RORE.

**Figure 2 cancers-14-01578-f002:**
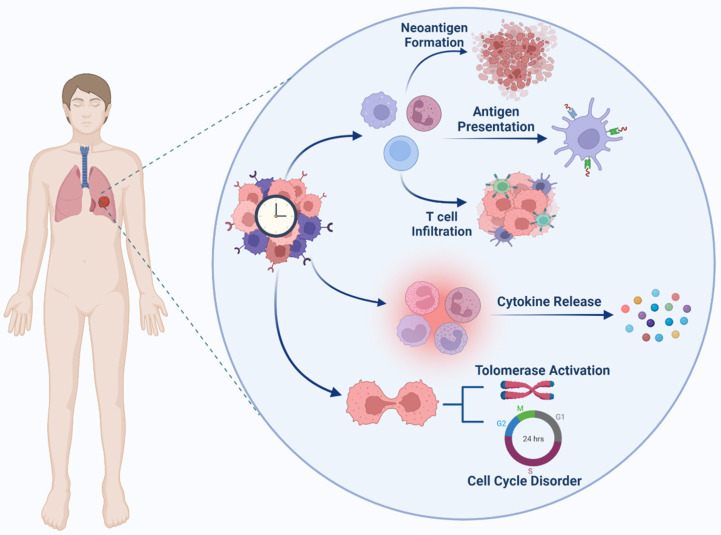
Effect of the circadian cycle on the main process of cancer such as cell cycle and immune behavior.

**Figure 3 cancers-14-01578-f003:**
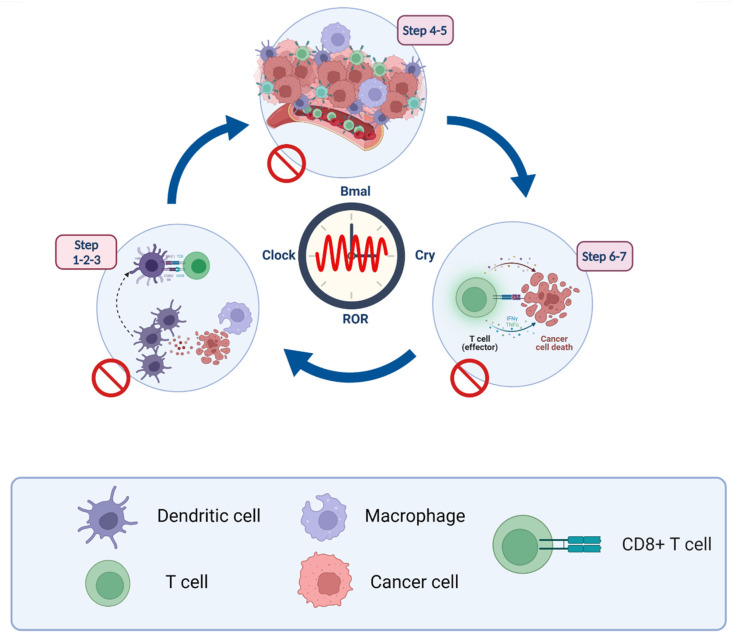
Cross talk between immunity cycle and circadian cycle genes; Recognizing, destroying, and releasing antigens from cancer cells are all part of this process. When NK cells are activated in tumor locations and come into direct contact with malignant cells, they kill them without the need for any kind of pre-exposure. In effector T cells, the circadian clock (ROR, PER1, CRY2, and BMAL1) adversely regulates PD-1 expression. CTLA-4 and PD-L1 are similarly negatively regulated by BMAL1 in effector T cells. IFN-γ, granzyme B, and perforin production by NK cells can be enhanced by PER-1 and BMAL1.

**Figure 4 cancers-14-01578-f004:**
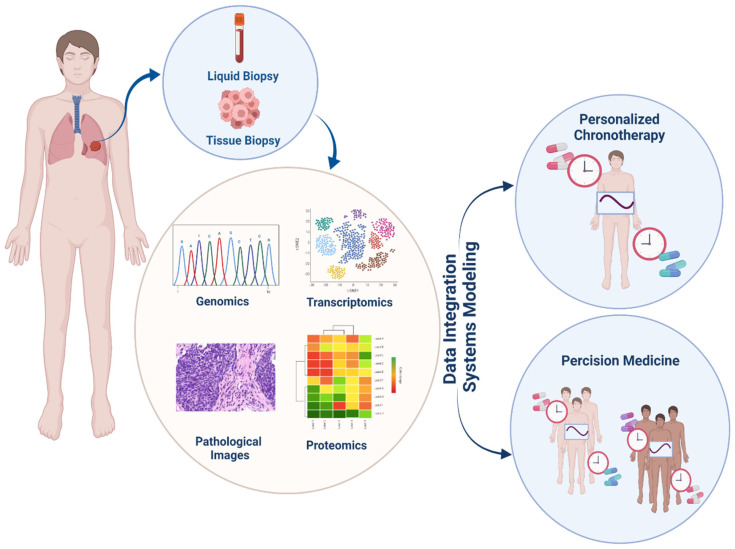
Our connection between the immune system and circadian rhythms could lead to a new area in cancer treatment. Using cellular and molecular data such as Omix data and pathology images, analyzing them, and establishing a network between these two cycles, determines the appropriate treatment for each person based on the administration time and the type of drug.

**Figure 5 cancers-14-01578-f005:**
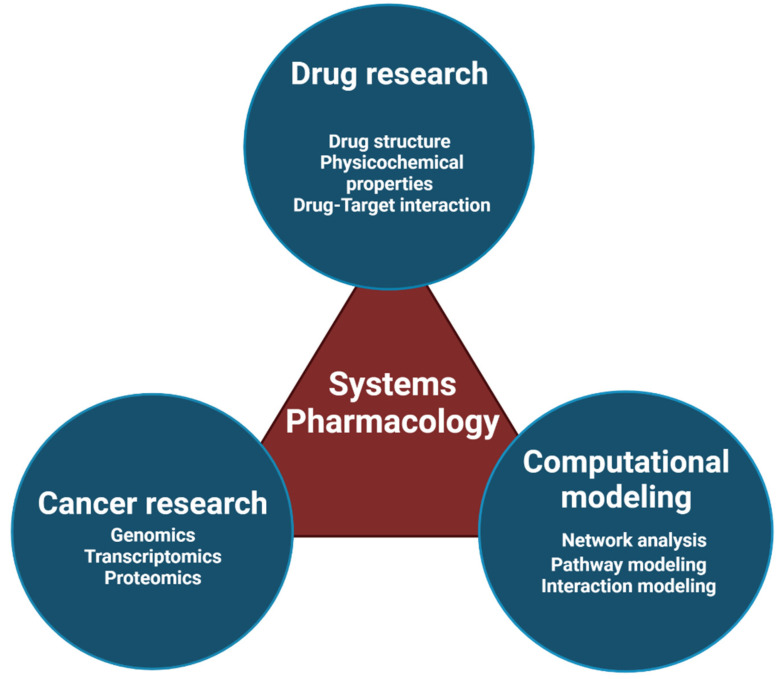
Systems pharmacology in cancer is a new field that has a significant role in the design, discovery, and repositioning of drugs due to the growth of basic cancer studies and the existence of accurate genetic and protein data and reliable data on drugs. These data, along with accurate bioinformatics models, can predict drug effects very well.

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
