# Peer review of "Circadian and Immunity Cycle Talk in Cancer Destination: From Biological Aspects to In Silico Analysis"

_cancers, 2022, doi:10.3390/cancers14061578_

Round 1

Reviewer 1 Report

The review describes the role of circadian rhythm in cancer and the interconnection with cancer immunity and treatment.

Several issues should be addressed:

  • Page 3. The authors state, “The circadian clock in mammalian cells is structured hierarchically”. As a review manuscript, it would be recommended to provide a schematic figure describing what is reported in the text of this paragraph. This would be very helpful for the reader.
  • The structure of the paragraph 3 “Role of circadian in cancer – Guilty or injured” should be reorganized. Subheadings for each short paragraph should be included (for example: genetic and genomic, DNA damage response pathway, Cell cycle, Tumor suppressor.

The part of inflammation should be moved to following paragraph 5, as that paragraph in specific for immunology.

Regarding the sentence “TGF signaling has been linked to components of the circadian clock in a variety of physiological situations”, please describe some examples.

In the Figure 2, add the p53 regulation.

Check the spelling of NF-kB.

  • The paragraph 4 sounds redundant with the paragraph 5. It should be reduced and the shorter version should be used as introduction for the paragraph 5.

In the sentence “Therefore, the inhibitory expression of CD4-positive T cells is a potential approach for the immunotherapy of lung cancer”, do the authors refer to Treg? It is not clear.

  • Similarly, paragraph 6 sounds redundant with the previous and not much relevant with the manuscript on circadian rhythm.
  • Paragraph 8, is too long and there is not much focus/application to biological clock. It should be reduced in length and more examples of modelling applied to the circadian rhythm and biological clock should be included.
  • Paragraph 9 seems not be relevant with the circadian rhythm. Please remove it or include examples that are pertinent with the scope of the manuscript.
  • The title of the review should be changed, as the authors included also other topics beside immunity (for example, therapy and modeling).

Reviewer 2 Report

In “Circadian and Immunity Cycle Talk: Vital Communication for Cancer Destination”, Mirian and colleagues review the connections between immune disruption in cancer and the molecular circadian clock.  After briefly introducing the circadian clock, they go over some of the non-immunologic connections between circadian rhythms and cancer.  They then spend several pages discussing how circadian rhythm influences different aspects of the immune system and the immune response, and how this might affect tumor biology, and bring up many interesting hypotheses.  Finally, in the last two sections, they introduce how computational biology and systems biology have informed immunotherapy in cancer.

Overall, this is a well-written and well-referenced review, though there are many grammatical mistakes and a few typos that should be corrected.  I have a few major comments and more minor comments below, but the one I want to highlight is that the Computational and Systems sections (sections 8 and 9) do not fit in well with the rest of the paper.  It is clear that the Authors are experts on these topics, and I appreciate their efforts to discuss them in detail.  However, there is very little in these sections that relates back to the main point of the Review, connections between circadian rhythms and immunity in cancer (there are just a few sentences in Section 8 about circadian rhythms, and none at all in Section 9).  The Authors should rewrite and / or shorten these sections to connect them better with the Review.  I look forward to reviewing a revised manuscript.

 Major points

  • The Title of the paper should be changed to better reflect that a large portion of it discusses computational approaches, otherwise readers will not know to access this Review for these topics.
  • For Section 3, it was hard to follow the constantly changing topics. Subsection headings about each molecular pathway might be helpful.
  • Section 3, page 4, section ii: It has never been experimentally shown that nutritional status of heme functionally affects REV-ERB activation in vivo, it is not clear that heme is ever limiting in cells. Please change this sentence to be more speculative.
  • Section 3, page 4, paragraph starting with “According to data from the Cancer Genome Atlas…”. References 51-53 do not test in any way whether rhythmicity is disrupted in cancer, rather they test whether circadian gene expression without regards to rhythmicity is disrupted.  Three papers have attempted to test rhythmicity of human tumor samples and should also be cited here: https://pubmed.ncbi.nlm.nih.gov/29404219/ , https://pubmed.ncbi.nlm.nih.gov/28439010/ , https://pubmed.ncbi.nlm.nih.gov/34726689/  
  • Sections 8 and 9 (Computational Biology and Systems Biology) do not fit in well with the rest of the Review and should be rewritten. These two sections spend several pages discussing technical matters in exhaustive detail, but there is very little attention paid to connections between circadian rhythms and the immune system, which is what the review is supposed to be about.  In fact, as far as I can tell, there is nothing about circadian rhythms in section 9.  These sections should be rewritten, or have introductions, to explain why the Authors are going to now discuss them and how they fit into the rest of the review.  Section 8 also has a lot of unnecessary technical detail and should be shortened.
  • Page 14, first full paragraph: the LYC drugs are specific agonists for RORγ, which has a poorly defined role in circadian rhythms (as opposed to RORα and RORβ, which are known to regulate BMAL1 levels). The Authors should discuss more clearly the specific role of RORy in circadian rhythms, or change the wording to mention that RORγ does not have a strong role in the molecular clock.

Minor points

  • In the abstract: “brilliant crosstalk” is not a scientific term. Please remove the word “brilliant”.
  • Page 2, first paragraph under Section 2: What are “exit genes”? Is this a typo?
  • Page 2, second paragraph under Section 2: NPAS2 is a paralogue of CLOCK, not of BMAL1, and cannot functionally replace loss of BMAL1. Please correct this sentence.
  • Page 2, second paragraph under Section 2: CLOCK-BMAL1 upregulation of PER and CRY is not a positive feedback loop, it is just normal transcription factor function. Please remove the term positive feedback loop from this paragraph.  The next paragraph correction describes how PER and CRY form a negative feedback loop to regulation CLOCK-BMAL1.
  • Please check all protein and gene names for consistency. For instance, through the paper, BMAL1 is named BMAL1, Bmal1, and BMAL-1, and PD-L1 is also named inconsistently.
  • On page 3, the paragraph about the SCN, melanopsin, and retinal ganglion cells has no connection to the rest of the review, and should be removed.
  • Also for section 3: for the pathways related to immune function on pages 5 and 6, these might fit better in Section 4, to help the reader follow the Author’s logic.
  • Section 3, page 4, paragraph starting with “According to data from the Cancer Genome Atlas…”. What is “genital clock”?  Is this a typo?  Or are the Authors referring to molecular clocks in ovary and testes?  Please clarify.
  • Pages 5-6: NFKB is not commonly referred to as NFB. Please correct to NFKB.
  • Page 6, second paragraph: which TGF are the Authors referring to? The TGF proteins have non-overlapping functions.
  • Page 9, second paragraph: the Authors begin to talk about tumor-associated macrophages (TAMs), but only discuss M1 macrophages, and then confusingly switch to talking about Th17 cells. Please address M2 macrophages and put the part about Th17 cells elsewhere, so this makes more sense for the reader.
  • Figure 4 would be helped by more labels in the Figure.
  • On Page 11, paragraph that starts with PDL-1, the Authors state, “PD-L1 is associated with an immune environment rich in CD8 T cells, production of Th-1 cytokines and chemical factors, as well as specific interferons and gene expression characteristics(118).” I am not sure this makes sense.  PD-L1 is associated with immunosuppression and exhaustion of CD8 cells.  Can the Authors better clarify what they were trying to say in this sentence?
  • Page 11, last paragraph, about metabolism: This paragraph is short and has no citations. Please remove it.
  • The Authors should work with an English-language editor to correct grammatical errors and inconsistencies throughout the paper, most notably in Sections 1-6, and the Abstract.

Round 2

Reviewer 1 Report

The authors addressed most of the questions and provided several changes to the previous version of the manuscript.

Author Response

We thank the reviewer for the positive comment.

Reviewer 2 Report

Thank you to the Authors for addressing my concerns and suggestions.  One very minor point to correct: 

Section 9 seems to have been removed or renumbered, the Review skips from 8.4 to 10.  Please renumber Section 10 to Section 9.

Author Response

We thank the reviewer for the positive comment. We have corrected the numbering of these sections.